# Hallmarks of the Tumour Microenvironment of Gliomas and Its Interaction with Emerging Immunotherapy Modalities

**DOI:** 10.3390/ijms241713215

**Published:** 2023-08-25

**Authors:** Christian A. Linares, Anjana Varghese, Aruni Ghose, Sayali D. Shinde, Sola Adeleke, Elisabet Sanchez, Matin Sheriff, Cyrus Chargari, Elie Rassy, Stergios Boussios

**Affiliations:** 1Guy’s Cancer Centre, Guy’s and St Thomas’ NHS Foundation Trust, London SE1 9RT, UK; christian.linares@nhs.net (C.A.L.); sola.adeleke@nhs.net (S.A.); 2Kent Oncology Centre, Maidstone and Tunbridge Wells NHS Trust, Hermitage Lane, Maidstone, Kent ME16 9QQ, UK; anjana.varghese@nhs.net; 3Department of Medical Oncology, Medway NHS Foundation Trust, Gillingham ME7 5NY, UK; aruni.ghose@nhs.net (A.G.); elisabet.sanchez@nhs.net (E.S.); matin.sheriff@nhs.net (M.S.); 4Barts Cancer Centre, Barts Health NHS Trust, London EC1A 7BE, UK; 5Mount Vernon Cancer Centre, East and North Hertfordshire NHS Trust, Northwood HA6 2RN, UK; 6Immuno-Oncology Clinical Network, UK; 7Centre for Tumour Biology, Barts Cancer Institute, Cancer Research UK Barts Centre, Queen Mary University of London, London EC1M 6BQ, UK; s.shinde@smd22.qmul.ac.uk; 8Faculty of Life Sciences & Medicine, School of Cancer & Pharmaceutical Sciences, King’s College London, Strand, London WC2R 2LS, UK; 9Department of Radiation Oncology, Pitié-Salpêtrière University Hospital, 75013 Paris, France; cyrus.chargari@aphp.fr; 10Department of Medical Oncology, Institut Gustave Roussy, 94805 Villejuif, France; elie.rassy@hotmail.com; 11Kent and Medway Medical School, University of Kent, Canterbury CT2 7LX, UK; 12AELIA Organization, 9th Km Thessaloniki–Thermi, 57001 Thessaloniki, Greece

**Keywords:** gliomas, tumour microenvironment, immunotherapy, immune checkpoint inhibitors, therapeutic cancer vaccines, oncolytic virotherapy, chimeric antigen receptors

## Abstract

Gliomas are aggressive, primary central nervous system tumours arising from glial cells. Glioblastomas are the most malignant. They are known for their poor prognosis or median overall survival. The current standard of care is overwhelmed by the heterogeneous, immunosuppressive tumour microenvironment promoting immune evasion and tumour proliferation. The advent of immunotherapy with its various modalities—immune checkpoint inhibitors, cancer vaccines, oncolytic viruses and chimeric antigen receptor T cells and NK cells—has shown promise. Clinical trials incorporating combination immunotherapies have overcome the microenvironment resistance and yielded promising survival and prognostic benefits. Rolling these new therapies out in the real-world scenario in a low-cost, high-throughput manner is the unmet need of the hour. These will have practice-changing implications to the glioma treatment landscape. Here, we review the immunobiological hallmarks of the TME of gliomas, how the TME evades immunotherapies and the work that is being conducted to overcome this interplay.

## 1. Introduction

The central nervous system (CNS) can be broadly divided into two cell types, neurons and glial cells, and gliomas originate from the glial cells, which include astrocytes, oligodendrocytes, ependymal cells and microglia. Gliomas comprise one of the most prevalent types of primary CNS tumours (PCNSTs), which are classified as Grade I to IV according to the World Health Organization (WHO) classification, taking into account histological, molecular and genomic features in their grading [1,2]. Glioblastoma, a WHO Grade IV glioma previously known as glioblastoma multiforme (GBM), is the commonest malignant PCNST, representing 49% of them and having an incidence of 3.23 per 100,000 of the population [1]. With a median overall survival (mOS) of 14.6 months and a 5-year survival rate of 5% despite surgical resection and adjuvant therapies, glioblastoma is certainly the centre of attention among PCNSTs [1,2,3].

Resistance to standard treatments for gliomas, such as the classic Stupp protocol [3], stems principally from the heterogeneity of the tumour microenvironment (TME), which is immunosuppressive and enables the evasion of the immune system, partially explaining the rapid disease progression [4]. Recently, novel treatment options are being investigated, such as immunotherapy. The aim of this review is to outline the immunobiological hallmarks of the TME of gliomas, how the TME evades immunotherapies and the work that is being carried out to overcome this interplay.

## 2. Hallmarks of the Tumour Microenvironment of Gliomas

(a)Cellular Armoury

Cancer is a disease that can arise in almost any tissue of the human body. Cancer arises when normal healthy cells transform into cancer cells that proliferate uncontrollably, leading to the formation of tumours. It is a leading cause of mortality worldwide, and the predicted risk of a cancer diagnosis is around 50% for individuals born post 1960 in the United Kingdom (UK) [5,6]. The hallmarks of cancer describe a set of characteristics acquired by healthy cells as they transform into neoplastic entities. The interaction between glioma cells and the TME is key for tumour proliferation and migration [7,8,9], and understanding the immunoregulatory entities and processes in the TME has uncovered many viable targets for developing antitumour strategies [10]. Glioma cells constitutively secrete C–C motif chemokine ligand 2 (CCL2), which converts T_h_2 lymphocytes into immunosuppressive T regulatory (T_reg_) cells and macrophages into the pro-neoplastic M2 phenotype [11]. In addition, glioma cells release C-X-C motif ligand 8 (CXCL8), which modifies the extracellular matrix through activating matrix metalloproteinases within in the TME [11,12,13]. Furthermore, through activation of tumour growth factor beta (TGF-β) and epidermal growth factor receptor (EGFR) signalling pathways, glioma cells can enhance their invasiveness [13].

A major part of the tumour bulk is comprised of immune cells such as tumour-associated myeloid cells (TAMCs) [13], subtypes of which include tumour-associated macrophages (TAMs), myeloid-derived suppressor cells (MDSCs), dendritic cells (DCs), neutrophils and microglia. Whilst not all myeloid cells are immunosuppressive, these TAMCs promote cancer growth directly by enhancing tumour cell proliferation and indirectly by generating an immunosuppressive microenvironment (Table 1) [13,14,15]. Microglia are present throughout the CNS and are key in regulating the cerebral immunological homeostasis [16]. Microglia are the resident CNS TAMs [17], which can secrete either immunosuppressive factors such as interleukin 10 (IL-10) and TGF-β or antitumour-stimulating cytokines such as IL-12 and TNF-α, according to the state of TME, whether ‘hot’ and highly infiltrated or ‘cold’ and poorly infiltrated [18].

Activated TAMs can exist in a spectrum of phenotypes, representing various functional states, such as tumour-suppressive M1 or immune-suppressive M2. Increased accumulation of TAMs with the M2 phenotype was correlated with a higher tumour grade and lower mOS or poor outcomes in recurrent glioblastoma [16,19]. TAMs have a high degree of plasticity and, therefore, can be reprogrammed, thus providing opportunities for their exploitation in treatment options.

DCs are ‘professional’ antigen-presenting cells (APCs) linking innate and adaptive immunity. They capture antigens and present them to T cells [15]. DC development comprises two distinct stages: immature and mature. Immature DCs predominantly reside in peripheral tissues, where they exhibit antigen-capturing abilities via phagocytosis and receptor-mediated endocytosis. In contrast, mature DCs are mainly found within lymph nodes and the spleen, displaying an enhanced antigen-presenting capacity with the elevated expression of co-stimulatory molecules like CD80 and CD86. These mature DCs effectively activate naive T cells, priming them to differentiate into effector T cells. DCs are usually present in the meninges and choroid plexus but are not seen within the normal brain parenchyma [13]. On the contrary, in a glioma-infiltrated brain, they are harboured within the parenchyma [20]. Some animal studies have demonstrated that these are recruited to the TME in a similar way to NK cells via chemokines CCL5 and XCL1 [20]. DCs are also essential in the activation of antitumour immune responses and interact with other immune cells through integration of the various TME signals [15]. They can secrete cytokines such as IL-12, leading to the increased recruitment of CD8^+^ T cells. However, they are still affected by TME immunosuppression, thus becoming regulatory DCs, which subsequently activate T_reg_ [21]. This leads to downregulation of CD8^+^ T-cell recruitment [22]. Increased IL-10 secretion by macrophages leads to reduced IL-12 production and results in the containing of DCs within the TME [20]. These mechanisms lead to inefficient DC differentiation and the formation of impaired DCs in immature cellular states, causing immunosuppressive conditioning of the TME [22]. DC-based vaccines against glioblastoma are presently under construction, and significant progress has been made over the past year [23,24].

The immune cells and the blood–brain barrier (BBB) are key to the TME’s adaptive alterations [9]. The BBB comprises a semipermeable membrane with endothelial cells, astrocyte foot processes and pericytes. This disconnects the brain from the peripheral immune system as evidenced by nil acute rejection of implanted grafts [25,26]. Naïve T cells cannot cross the BBB, but activated T cells can [25]. The BBB, thus, tightly regulates leukocyte entry into the brain parenchyma, due to which gliomas experience an overall decreased immune surveillance as compared to other tumours [16]. Furthermore, this tight regulation accounts for the poor therapeutic effectiveness of lipophobic intravenous treatments. In gliomas, the tumour physically distorts the BBB and induces inflammation, which then causes the surrounding blood vessels to become leaky and compromised [25]. The inadequate blood flow creates hypoxic regions within the tumour due to insufficient oxygen delivery, and these areas then attract macrophages, which further enhances the tumourigenicity of gliomas [21].

(b) The Lymphocytic Milieu

Physiologically, the cytokine environment of the CNS is regulated towards helper T cell lymphocytes (T_h_2) to shield the brain against inflammatory destruction [21]. Gliomas exploit this response by enhancing tumour-infiltrating lymphocyte (TIL) production of T_h_2 cytokines [13,14].

Regulatory T cell (T_reg_) suppress the activity of effector T cells and DCs. Whilst no T_reg_ are found in normal brain tissue, increased numbers of T_reg_ cells are seen in a glioma-infiltrated brain. This offers the key ability of a glioma to evade the immune system, as will be discussed in onward sections [13]. These cells are recruited to the TME by the secretion of chemokines such as CCL2 and CXCL12 by glioma cells. The number of T_reg_ present is linked to the location and grade of the tumour [13,21]. They induce compromised APCs, which have decreased ability to activate tumour reactive T cells [21]. In addition, T_reg_ secrete factors such as IL-10 and TGF-β, which inhibit the activity of other immune cells [15]. M2-phenotype macrophages and T_reg_ infiltrating the glioblastoma also leads to suppression of T-cell function [8]. A study showed that this concept was successful in treating ovarian cancer [27].

Natural killer (NK) cells are CD3^−^, CD56^+^ and CD16^+^ innate lymphocytes that induce cytotoxic apoptosis in cells, therefore playing a vital role in the immune response [18]. NK cells are characterised by the expression of specific receptors, including killer cell immunoglobulin-like receptors (KIR) and killer cell lectin-like receptors (KLR), also known as killer activation receptors (KAR). NK cells can recognise virally infected or malignant cells by their absent major histocompatibility complex (MHC) class I and cause apoptosis by exhibiting a combination of inhibitory as well as stimulatory receptors [13,14,15]. Studies have shown that NK-cell deficiencies were correlated with an increased incidence of certain cancers, including glioblastoma [28,29]. Furthermore, glioblastoma expresses human leukocyte antigen G (HLA-G), which further limits the action of NK cells, providing protection from NK-cell-mediated death [18]. HLA-G interacts with inhibitory receptors on NK cells, suppressing cytotoxicity and inhibiting the ability to recognise and attack tumour cells. This immune evasion mechanism provides protection from NK-cell-mediated death, contributing to tumour resistance and disease progression. NK-cell activity is also hindered by MDSCs through the production of arginase and reactive oxygen species (ROS) [8].

(c) Immunosuppressive Factors and Immune Evasion

The glioma microenvironment secretes a variety of immunosuppressive factors, such as TGF-β2, prostaglandin E2 (PGE2), IL-1, IL-10 and fibrinogen-like protein 2 (FGL2). These factors collectively further suppress effector T cell activity [13]. In addition, T_reg_ cells and MDSCs further prevent the normal NK-cell- and cytotoxic T lymphocyte (CTL)-mediated cytotoxic reactions [15,16]. TGF-β1 and IL-10 skew TAMCs toward the immunosuppressive M2 phenotype, which then along with T_reg_ secrete further TGF-β1 and IL-10, hence suppressing the immune system [14]. This immunosuppressive phenotype enables aggressive tumour proliferation and invasion, while inhibiting the normal antitumour immune responses [15].

Gliomas also express programmed death-ligand 1 (PD-L1), which is the primary ligand of programmed cell death protein 1 (PD-1), resulting in T-cell exhaustion and anergy [21].

Chronic antigenic stimulation in the TME induces T-cell exhaustion, characterised by impaired cytokine production, cytotoxicity and proliferation. This exhaustion is mediated by immune checkpoint molecules such as PD-1 and cytotoxic T-lymphocyte-associated protein 4 (CTLA-4). Immune checkpoint inhibitors (ICIs) targeting these pathways have revolutionised cancer treatment by reinvigorating exhausted T cells.

T-cell anergy is a common tolerance mechanism in which T cells are functionally inactivated, thus unable to coordinate a response after encountering an antigen, but remain in a prolonged, hyporesponsive state. Both types of anergies, i.e., clonal/in vitro and adaptive/in vivo, are seen in glioblastoma [30]. In clonal anergy, ineffective Ras/mitogen-activated protein kinase (Ras/MAPK) pathway activation and defective co-stimulation leads to impaired T-cell activation. Adaptive anergy, on the other hand, has persistent low-level antigen stimulation causing T-cell desensitisation, which leads to defective nuclear factor kappa-light-chain-enhancer of activated B cells (NF-κB), decreased IL-2 release and impaired T-cell amplification [14,30].

The ability of glioma cells to evade the immune system is key in allowing them to proliferate. This mechanism depends on the anatomical site of the tumour within the CNS and the intrinsic cell-to-cell interactions among the tumour and the immune cells [13,14,15]. One of the most effective ways in which glioma cells cause immunosuppression is by reducing the overall recruitment of immune cells, while increasing the recruitment of microglial cells [16]. These microglia appear like immature APCs, lacking the ability to provide T-cell-mediated immunity. In addition, gliomas release immunosuppressant cytokines such as TGF-β, IL-10 and cyclooxygenase 2 (COX-2), while simultaneously inhibiting signal transducer and activator of transcription 3 (STAT3), thus enhancing the immunosuppressive microenvironment [21]. Hypoxia within TME due to impaired blood vessels and greater usage of oxygen by tumour cells results in the activation of the immunosuppressive STAT3 pathway. This STAT3 pathway leads to the creation of hypoxia-inducible factor-1 alpha (HIF-1α), the stimulation of T_reg_ cells and the synthesis of vascular endothelial growth factor (VEGF), and VEGF then further alters the vasculature and inhibits DC development, antigen presentation and T-cell infiltration into tumours [22].

Antigen recognition following presentation is essential for T-cell-mediated immunity, and this relies on the expression of MHC molecules [7]. Invading gliomas downregulate the expression of MHC proteins and costimulatory molecules such as CD80 and CD86 on their surface, leading to reduced immune recognition and the activation of cytotoxic T cells (CTLs) [8,14]. As mentioned above, the IL-10 and TGF-β enriched immunosuppressive TME of gliomas leads to loss of MHC expression on microglia [21]. Furthermore, reduced expression of MHC class I proteins was also present in glioma stem cells, in turn adding to T-cell-mediated immunity resistance and leading to increased tumour proliferation [16].

The blockage of chemotactic agents with antibodies or therapeutic drugs supresses the recruitment of suppressor cells. TGF-β is key in the development of T_reg_ cells and is upregulated in gliomas [14,21]. Antisense phosphorothioate oligodeoxynucleotide trabedersen (AP 12009) has been shown to successfully inhibit TGF-β expression in vitro, and in animal models the inhibition of TGF beta pathways among gliomas helped to re-establish immune surveillance [31]. Thus, inhibiting the cytokine production of glioma cells decreases their ability to proliferate, thus reducing their capacity to recruit immunosuppressive cells [32].

## 3. Immunotherapy and the Interplay with the Tumour Microenvironment

(a)Immunotherapy Landscape in Glioma

As alluded to previously, the standard of care (SOC) for glioblastoma is surgical resection in with adjuvant radiotherapy and chemotherapy, mainly with temozolomide (TMZ), as per the Stupp protocol [3]. A high-dose steroid, most commonly dexamethasone, is also administered to reduce vasogenic cerebral oedema, and all of these treatments further suppress the immune system. For example, pancytopenia and TMZ-induced lymphopenia are common side effects. Even a reduced dose of dexamethasone can lead to a ‘colder’ TME, with fewer infiltrating lymphocytes, reduced microglial trafficking and the blunted release of proinflammatory cytokines, posing a challenge for clinical oncologists to weigh the benefit of reducing vasogenic oedema against the immunosuppressive side effects of steroids and consider using the lowest dose possible [21,33]. As the SOC alone is unlikely to improve prognosis or survival in these diseases, immunotherapy has emerged as a promising avenue for the treatment of gliomas, possibly in combination with SOC [34]. The focus of glioma immunotherapy research has centred on four approaches: ICIs, chimeric antigen receptor (CAR) T and NK cells, cancer vaccines and oncolytic viruses (OVs). Some of the more prominent types of immunotherapies are described in Table 2.

(b) Immune Checkpoint Inhibitors

T-cell activity is mediated through integrating both stimulatory and inhibitory signals, collectively termed immune checkpoints, which function to prevent the immune system from attacking one’s own cells. However, some cancer cells can manipulate these checkpoints within the TME to evade the immune system, allowing neoplastic proliferation. ICIs are a ground-breaking class of humanised immunoglobulin G (IgG) monoclonal antibodies (mAbs) that have revolutionised cancer treatment in the last decade by enabling the immune system to recognise and attack cancer cells effectively [35].

There are three principal types of ICIs which have been approved for clinical use. Nivolumab, pembrolizumab and cemiplimab are anti-PD-1 IgG4 mAbs that target the inhibitor receptor PD-1 on activated T cells, NK cells, B cells, macrophages and several subsets of DCs, thus activating immune cells by interfering with the CD28-costimulatory signalling pathway. Atezolizumab, durvalumab and avelumab are anti-PD-L1 IgG1 mAbs that target PD-L1, the main ligand of PD-1, along with PD-L2, which is constitutively expressed on APCs within the TME as well as a wide range of tumours, such as lung, breast, and melanoma, thereby disinhibiting the migration and activation of T cells to seek and destroy PD-L1-expressing cancer cells [39,40]. Ipilimumab is an anti-CTLA-4 IgG1 mAb that targets CTLA-4, which normally governs the amplitude of T-cell activation, thereby blocking the normally immunosuppressive effect of the CD28-costimulatory signalling pathway of T cells and increasing their activation and proliferation (Figure 1) [35]. The immune-related adverse events (irAEs) associated with ICIs tend to be different from the side effects classically associated with chemotherapies and are quite pleomorphic in their manifestations, with combinations of ICIs more likely to produce higher-grade irAEs and at a much more accelerated rate than monotherapies [41]. However, severe and refractory irAEs are generally manageable with well established guidelines on targeted immunosuppression depending on the irAE [42].

Ipilimumab was first approved to treat melanoma, but when combined with nivolumab it can also be used to treat advanced renal cell carcinoma (RCC), microsatellite instability/deficient mismatch repair (MSI-H/dMMR) metastatic colorectal carcinoma (mCRC), malignant pleural mesothelioma (MPM), non-small-cell lung carcinoma (NSCLC) and hepatocellular carcinoma (HCC) [35,43]. In the setting of recurrent glioblastoma, monotherapy with PD-1 blockade yielded a mOS comparable with that of bevacizumab [44], an anti-IgG1 mAb targeted against VEGF-A known to prolong median progression-free survival (mPFS) [45]. In mice with glioblastoma, combining stereotactic radiotherapy with PD-1 blockade resulted in 75% pathologic complete response by activating macrophages, highlighting a novel immunologic mechanism underlying the interaction between radiotherapy and ICIs, though an international phase 3 trial demonstrated longer mOS from TMZ with radiotherapy than nivolumab with radiotherapy, leaving the SOC for glioblastoma unchanged as of now [46,47].

Gliomas are one of many types of cancers that manipulate these pathways to inactivate T cells within the TME. As described, PD-1 is an inhibitory membrane protein present on activated T cells to dampen the immune response. It is activated by ligands PD-L1 and PD-L2 found on tumour cells and infiltrating immune cells, and an increased presence of PD-L1 was associated with a higher grade of glioma and poorer prognosis in patients [48,49]. Following the failure of ICI monotherapy, attention is now on combining therapies to simultaneously block multiple drivers of T cell exhaustion, such as with bispecific antibodies targeting TGF-β, PD-L1 and CD27, or with existing elements of SOC like RT and TMZ, or targeting CCR4 to reduce T_reg_ migration and disrupting immunosuppressive stromal components of the TME [50]. Nivolumab alone did not demonstrate any prognostic benefit for relapsed glioblastoma; however, it is presently being explored as adjunct to radiotherapy and/or TMZ in newly diagnosed glioblastoma [51]. Two recent studies have demonstrated that anti-PD-1 mAbs in combination with surgical resection leads to significantly improved mOS in glioblastoma as compared to adjuvant therapy alone [51,52]. Other studies using different mAbs have also found similar results. However, larger-scale trials are required to robustly prove the efficacy of the neoadjuvant approach. Moreover, using combinations of different and unconventional immunotherapies might be a potential management approach to overcome the heterogeneous and highly immunosuppressive nature of gliomas.

(c) Therapeutic Cancer Vaccines

Cancer vaccines can be preventive or therapeutic. Preventive ones such as those targeting human papillomavirus (HPV) and hepatitis B virus (HBV), have been successful in reducing the risk of cervical and hepatocellular cancer respectively [53,54]. In contrast, therapeutic cancer vaccines aim to stimulate the immune system to recognise and attack existing cancer cells (Figure 1) [36]. These are an example of active immunotherapy, as they work predominantly through the activation of CTLs via the presentation of tumour-associated antigens (TAAs) by APCs such as DCs. DC-based vaccines involve extracting DCs and exposing them to TAAs before being reintroduced into the patient’s body, whereas tumour cell-based vaccines utilise whole tumour cells or specific antigens from the cancer cells to stimulate the immune system. They can be administered in numerous ways. The first method involves the administration of TAAs, which are then presented to T cells by APCs to invoke an immune response. The second way involves priming autologous DCs ex vivo with the patient’s TAAs and then re-administering these cells intradermally to the patient, a technique termed DC vaccination [24].

In 1990, Bacillus Calmette-Guérin (BCG) became the first ever immunotherapy to be approved for use and the first therapeutic cancer vaccine, licensed for use in superficial early stage bladder cancer [55]. In 2010, Sipuleucel-T, a DC-based vaccine, was approved after being shown to confer a significant survival advantage to patients with asymptomatic hormone-refractory prostate cancer [56]. In 2022, a study found that adding autologous tumour lysate-loaded DC vaccine (DCVax-L) to SOC resulted in a significant extension of OS for patients with both newly diagnosed and recurrent glioblastoma, with an even greater relative survival benefit observed among patients who would have generally fared worse with SOC [24]. Whilst DCVax-L is not yet approved by the Food and Drug Administration (FDA) in the United States or the Medicines and Healthcare products Regulatory Agency (MHRA) in the UK, it has recently been made available for private use in the UK (Northwest Biotherapeutics 2017) [57], and the National Institutes of Health and Care Excellence (NICE) are conducting a technology appraisal of the clinical and cost effectiveness of DCVax-L for newly diagnosed glioblastoma [58].

However, there are several challenges in developing effective treatments, namely the need for the better identification of TAAs, strategies to overcome immune evasion and optimisation of vaccine delivery and adjuvant use. Additionally, the development of combinatorial immunotherapies synergistic with cancer vaccines, such as ICIs or targeted therapies, may lead to more durable responses. As research in these areas continues, cancer vaccines may become an essential tool in the fight against cancer [36].

(d) Chimeric Antigen Receptor T and NK Cells

A chimeric antigen receptor (CAR) is a synthetic receptor engineered to redirect immune cells, such as T cells and NK cells, to target specific antigens on the surface of cancer cells. This adoptive approach involves the genetic modification of patient-derived T cells to express CARs to recognise specific TAAs. These engineered T cells are then infused back into the patient, where they can target and kill cancer cells. CAR T-cell therapy has shown success in haematological malignancies, specifically diffuse large B-cell lymphoma (DLBCL) [59] and B-cell acute lymphoblastic leukaemia (B-ALL) [60].

Clinical trials of CAR T-cell therapy for gliomas have primarily focused on targeting TAAs such as IL-13 receptor alpha 2 (IL-13Rα2) [37], EGFR variant III (EGFRvIII) [61,62] and human EGFR 2 (HER2) [63]. EGFRvIII, for instance, is a tumour-specific mutant of EGFR found in a subset of glioblastoma and has been associated with poor prognosis [64]. However, a phase 1 trial of EGFRvIII targeted CAR T cells demonstrated only transient reductions in tumour size and EGFRvIII expression in select patients (Figure 2) [62].

Translating to the glioma setting is challenging due to TAA heterogeneity, the immunosuppressive microenvironment and the BBB [65,66]. The heterogenous expression of TAAs can result in the escape of antigen-negative tumour cells, leading to relapse [67]. Strategies to target multiple antigens simultaneously using dual or multi-antigen targeting of CAR T cells, which could avoid antigen escape within the TME are in the pipeline [63]. The immunosuppressive glioma TME consisting of T_reg_ cells, MDSCs and TAMs, as well as inhibitory molecules like PD-L1, can impair the function and persistence of CAR-T [67]. Incorporating a cell-intrinsic PD-1 checkpoint blockade within CAR T cells by engineering the expression of the PD-1-dominant negative receptor (DNR), a decoy receptor that binds PD-L1 on tumour cells, is a promising strategy as the co-transduction of PD1-DNR with a CAR has been shown to enhance T-cell functional persistence and T-cell resistance to tumour-mediated T-cell inhibition, thus disrupting the inhibitory action of this TME element and maintaining T-cell activation [68]. Another strategy is combining CAR T-cell therapy with cell-extrinsic PD-1 blockade with ICIs such as nivolumab [66]. However, this approach is likely to come with significant safety concerns, noting particularly the famous case of a HER2-specific CAR T cell causing respiratory failure and death in a patient lung- and liver-metastatic HER2^+^ breast carcinoma, revealing a potential ‘on-target, off-tumour’ effect of CAR T cells directed at a target also found in normal tissue [69].

The BBB can physically limit the trafficking of systemically infused CAR T cells into the brain and the tumour site [65]. Strategies to improve CAR T-cell infiltration across this anatomical barrier into the CNS include direct intracranial administration, such as intratumoural or intraventricular infusion [70,71]. Crossing the physiologic BBB is then dependent on appropriate matched expression of adhesion molecules and chemokine receptors, namely CXCR3 and CCR5, to facilitate endothelial adhesion and translocation. However, these tumour-bound ligands are typically expressed in very low quantities. So, another strategy being explored is the engineering of CAR T cells that express better-matched chemokine receptors [72]. Once CAR T cells enter the brain parenchyma, they encounter the immunosuppressive TME, which induces T-cell exhaustion and apoptosis as previously described. To recruit T_reg_ cells, gliomas overproduce factors like indoleamine 2,3-dioxygenase 1 (IDO-1), and glioma stem cell (GSC)-derived pericytes secrete CCL5, whereas cerebral stromal cells produce immunosuppressive cytokines, namely TGF-β and IL-10 [50].

CAR NK-cell therapy is another potential therapeutic avenue for glioblastoma. Unlike T cells, NK cells, as mentioned before, are part of the innate immune system [73]. They directly recognise and eliminate cancer cells without prior antigen experience via an antigen-independent mechanism [74]. Activated NK cells release various cytotoxic molecules like perforin, granzymes and IFN-γ, which induce tumour apoptosis. Another mechanism is FcγRIIIA/CD16a-mediated antibody-dependent cellular cytotoxicity (ADCC) [75]. Moreover, NK cells also regulate and activate the adaptive immune response through molecular crosstalk with DCs, enhancing tumour antigen presentation to modulate T-cell-mediated immunity antitumour responses. By switching from conventional CAR T-cell to NK signalling domains, CAR NK cells exhibit improved tumour-killing function. The targets being explored for CAR NK cells in glioblastoma are like those of CAR T-cell therapies [73].

Initial trials of NK-cell therapy for glioblastoma have focused on autologous approaches, utilising ex-vivo-expanded activated NK cells derived from the patients’ peripheral blood mononuclear cells (PBMC). These autologous adoptive therapies have demonstrated safety and shown durable responses to recurrent glioblastoma [76]. To note is the limited cytotoxicity of autologous NK cells against glioblastoma. In contrast, allogeneic NK cells sourced from healthy donors are highly cytotoxic and have minimal risk of graft-versus-host disease (GvHD) [77]. Therefore, allogeneic therapy holds promise for generating off-the-shelf cellular therapy products, bypassing inhibitory signals, and simplifying manufacturing processes. Current studies have demonstrated their safety and efficacy in haematological malignancies, along with some success in the solid tumour landscape [78].

Whilst preclinical models have demonstrated the efficacy of CAR-NK in orthotopic mouse xenograft models, several barriers persist [75]. Glioblastomas restrict NK-cell infiltration and downregulate target antigens. As previously described, the TME releases inhibitory cytokines and chemokines such as TGF-β to evade NK-cell-mediated oncolysis. Combining NK cells with TGF-β inhibitors or other agents, like cationic supramolecular inhibitors and ICIs, shows potential in overcoming these obstacles [79]. However, technical challenges in CAR-NK development, large-scale manufacturing, and need to create bespoke molecules remain major limiting factors for all types of CAR therapies. This warrants the optimisation of gene-modification and -expansion methods for successful clinical trials of CAR NK-cell and T-cell therapies for glioblastoma [73].

(e) Oncolytic Virotherapy

Oncolytic viruses (OVs) represent a novel treatment strategy in cancer immunotherapy, referred to as oncolytic virotherapy (OVT), due to their dual mechanisms of action: directly lysing cancer cells, and modulating the TME to stimulate antitumour responses. OVs selectively replicate within cancer cells leading to their apoptotic destruction, known as oncolysis [80]. As OV-infected cancer cells die, they release tumour antigens which are taken up by APCs and presented to T cells, educating them to identify and kill specific cancer cells, thus promoting an adaptive immune response [81]. Oncolysis leads to the release of damage-associated molecular patterns (DAMPs) and pro-inflammatory cytokines. These further stimulate the immune system, converting the ‘cold’ immunosuppressive TME, like that of glioblastoma, into a ‘hot’ immunostimulatory one, like that of melanoma, lending OVT, facilitating synergism with other immunotherapies like ICIs and CAR-T [38]. OVs can also be genetically engineered to express immunomodulatory molecules boosting the immune response i.e., promoting drug activation or directly inhibiting tumour growth. Currently, seven OV platforms are under investigation in neuro-oncology. DNA viruses include herpes simplex virus 1 (HSV-1), adenovirus (AdV), vaccinia virus and parvovirus, whereas RNA viruses include poliovirus (PV), reovirus and measles virus. Each platform has its pros and cons and different modes of delivery [81].

In 2022, teserpaturev became the world’s first OVT approved for glioma based on the landmark Japanese single-arm phase 2 trial. A third-generation oncolytic HSV-1 called G47Δ was delivered intratumorally via a stereotactic neurosurgical procedure to 19 patients with either residual or recurrent glioblastoma. The primary endpoint of 1-year survival rate after G47Δ initiation was 84.2%, which is a substantial improvement from 30%. The mOS was 20.2 months after G47Δ initiation and 28.8 months from the initial surgery, which is significantly longer than standard mOS of under a year with existing therapies. The best overall response in 2 years was a partial response in 1 patient and stable disease in 18 patients. On MRI, oncolysis was suggested by the characteristic enlargement and contrast clearing within the target lesion after each repeated G47Δ administration. Tumour biopsies showed increasing numbers of tumour infiltrating CD4^+^ and CD8^+^ lymphocytes, indicating an immune response, as well as persistently low numbers of FOXP3^+^ T_reg_, indicating decreased immune suppression within the TME (Figure 3). Adverse reactions to teserpaturev mainly comprised of symptoms suggestive of mild viral illness, likely related to the immune system attempting to eradicate such unnaturally large load of virus. However, no dose-limiting toxicity was observed, and indeed the concept of maximum tolerated dose as applied to the development of chemotherapies may not be so relevant in the development of OVTs [82].

However, several challenges that need to be addressed for OVT to be adopted as a real-world modality. These include the immune potential to neutralise OVs prior to tumour infection, ability of OVs to infect and kill all types of cancer cells, and ensuring the safety of using live viruses. Ongoing strategies include the combination of OVT with standard therapies [38,81]. RT can enhance OV replication in tumour cells by altering gene expression, for instance, by upregulating human transcription factor Y-box binding protein 1 (YB-1) in the glioblastoma cell nuclei to upregulate the replication of oncolytic AdV dl520 [83]. Another recent phase 1 trial of AdV-tk, an oncolytic AdV engineered to express HSV thymidine kinase (HSV-tk), demonstrated a safe RT and OVT combination in paediatric high-grade gliomas [84]. OVT is also showing promise for overcoming TMZ resistance, i.e., the oncolytic paramyxovirus Newcastle disease virus (NDV) inhibits the Akt signalling pathway and enhances the antitumour effect of TMZ [85]. Another example is the combination of oncolytic AdV DNX-2401 with TMZ, which greatly enhances the CD8^+^ recognition of glioblastoma cells [86].

The combination of OVT with other immunotherapy modalities is particularly attractive as it offers direct glioma TME immunomodulation, which is the principal limiting factor. Looking at ICIs, monotherapies yielded lacklustre results, and combination therapies resulted in severe adverse reactions, especially with anti-PD1 and anti-CTLA-4 mAbs together [38]. However, OVs can increase the effectiveness of other immunotherapy modalities in glioblastoma by essentially reprogramming the TME to enhance the antitumour properties of the other immunotherapies and allow synergism [81,87]. OVs were shown to induce the upregulation of PD-1 on T cells and PD-L1 on tumour cells, thereby increasing the sensitivity of gliomas to ICIs [88]. Also, a phase 2 trial of oncolytic AdV DNX-2401 with anti-PD1 pembrolizumab achieved a median OS of 12.5 months [89].

The combination of OVs with CAR-T and CAR-NK have also shown promising results in the face of poor penetration when used alone and the highly immunosuppressive glioma TME. For example, loading a CAR-T cell with tumour-specific mAbs can help overcome the on-target/off-tumour cross-reactivity of some CAR-T cells with both glioma and normal cells, such as in Lp2 CAR-T cells loaded with LpMab-2 to target podoplanin (PDPN)-expressing glioma cells whilst sparing PDPN-expressing normal cells, when used with G47Δ [90]. Oncolytic HSV-1 (oHSV-1) enhanced the therapeutic efficacy of CD70-targeted CAR-T by increasing intratumoural T and NK-cell infiltration and IFN-γ release within the TME of glioblastoma [91]. When used in combination with B7-H3 CAR-T, an oncolytic AdV loaded with CXCL11, called oAds-CXCL11, led to the increased infiltration of CD8^+^, NKs and M1-polarised macrophages, as well as decreased levels of MDSCs, T_reg_ and M2-polarised macrophages, when compared to B7-H3 CAR-T alone in mice [87]. The combination of OV-IL15C, an oncolytic HSV-1 that expresses IL15/IL15Rα fusion protein, and off-the-shelf EGFR-CAR-NK showed a synergism in inhibiting tumour growth and improving survival in mice compared to using either as monotherapy. This was associated with higher levels of NK and CD8^+^ infiltration and activation within the brain, as well as the increased persistence of CAR-NK. These findings were noted in an immunocompetent model [92]. These combinations represent a significant frontier in the development of immunotherapies targeting gliomas [81].

(f) Future Directions

There is currently quite a lot of work being conducted to investigate ways to overcome the immunosuppressive TME to enhance existing and emerging treatment strategies and pave the way for new, undiscovered approaches, with a number of promising targets on the horizon [93]. The Krebs cycle metabolite itaconate, secreted by MDSCs through the activity of immune-responsive gene 1 (IRG1), is taken up by CD8^+^ T cells to suppress the proliferation of CD8^+^ T cells and cytokine production, and the deletion of IRG1 in mice has been shown to enhance the antitumour activity of the anti-PD-1 blockade [94]. Oncostreams, which are fascicles of aligned spindle-like cells that facilitate the intratumoural distribution of tumour cells, depend on the overexpression of collagen, alpha 1, type I (COL1A1), the gene that encodes the major component of type I collagen; so, the inhibition of COL1A1 has been shown to reprogramme the malignant behaviour of gliomas and alter the TME, highlighting oncostreams as yet another high-value target which can possibly be exploited by immunotherapies for gliomas [95].

## 4. Conclusions

Gliomas including glioblastomas are notorious for poor prognosis. Existing standard-of-care regimens are neither highly effective nor offer a lucrative survival benefit. The TME has a challenging heterogenous, immunosuppressive milieu facilitating immune evasion and tumour proliferation. Immunotherapy modalities including ICIs, therapeutic cancer vaccines, OVT and CAR T-cell and NK-cell therapies are emerging gamechangers. Combination therapies using these are increasingly being translated into the glioma setting as TME shortcomings are being overcome. The clinical trials in the pipeline over the last decade have shown promising results in efficacy and survival outcomes. Rolling out these multimodal immunomodulatory cocktail therapies in the real world is an unmet need of the hour. If executed in a low-cost, high-throughput manner, landscape changes in the mainstay of glioma therapy are expected.

## Figures and Tables

**Figure 1 ijms-24-13215-f001:**
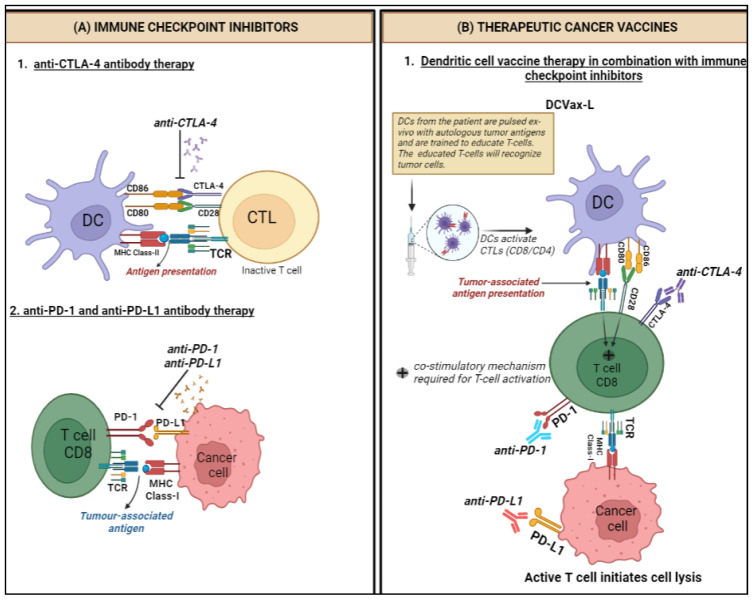
Mechanism of action of immune checkpoint inhibitors and therapeutic cancer vaccines for glioblastoma. (**A**) Immune checkpoint inhibitors (ICIs): (1) Co-stimulatory activation of T cells is achieved through the TCR and CD28 receptors. CD28 binds to the CD80/86 ligands on the DCs, but the CTLA-4 receptor competes with CD28 for binding to CD80/86 and leads to the inactivation of T cells. Anti-CTLA-4 mAbs are developed which bind to the CTLA-4 receptors. (2) Cancer cells overexpress PD-L1 receptors which bind to the PD-1 ligands on the T cells, leading to T-cell inactivation. Anti-PD-1 and anti-PD-L1 mAbs bind to PD-1 and PD-L1. (**B**) Therapeutic cancer vaccines: DCs derived from the patient’s peripheral blood monocytes are pulsed ex vivo with tumour lysate and are trained to recognise T cells. The educated T cells recognise tumour antigens and initiate cell lysis. DCVax-L is used to treat brain tumours in combination with ICIs. Dendritic cells (DCs), T cell receptor (TCR), cytotoxic T lymphocytes (CTLs), major histocompatibility complex (MHC). Created with BioRender.com.

**Figure 2 ijms-24-13215-f002:**
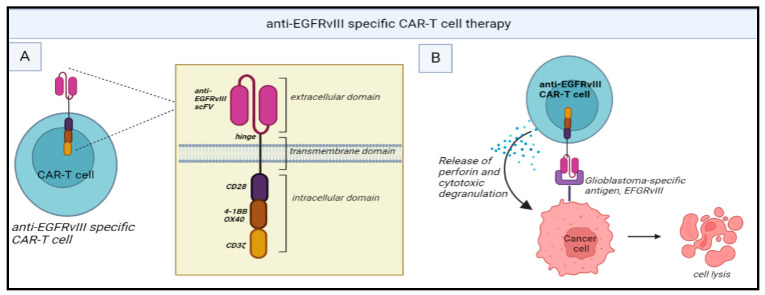
(**A**) Structure of anti-EGFRvIII-specific CAR T cells: It consists of a single-chain fragment variable (scFv) for anti-EGFRvIII mAbs along with CD3ζ (signaling domain for TCR). The intracellular domain brings about T cell activation. (**B**) Mechanism of action: anti-EGFRvIII specific CAR T cells recognize EGFRvIII antigens present in the glioblastoma cells and this attachment leads to the release of perforin leading to cytotoxic degranulation. Chimeric antigen receptor (CAR), Epidermal growth factor receptor (EGFR), T cell receptor (TCR). Created with BioRender.com.

**Figure 3 ijms-24-13215-f003:**
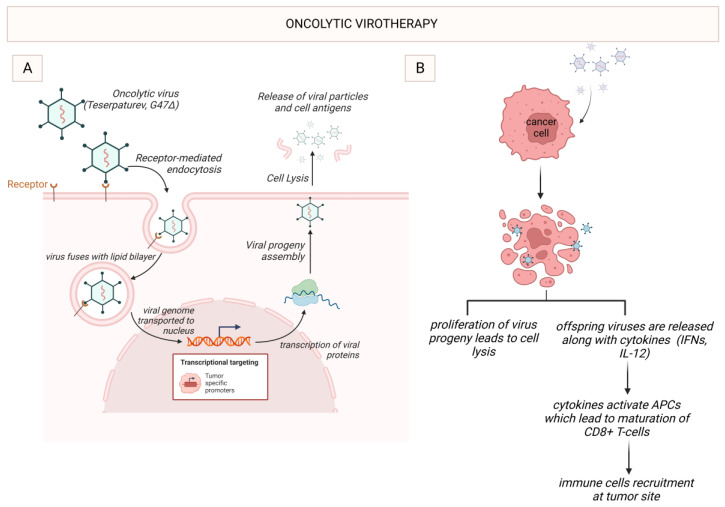
Oncolytic virotherapy with G47Δ. (**A**) G47Δ, an oncolytic HSV-1, enters the tumour cell through receptor-mediated endocytosis. Once inside the cell, it undergoes viral replication, leading to the release of virus progeny. (**B**) Once G47Δ enters the tumour cell, cell lysis leads to proliferation of the viral progeny and the release of offspring viruses and cytokines such as IFNs. This activates APCs such as DCs, which further mature the cytotoxic T lymphocytes, such as CD8^+^ T cells, leading to immune stimulation. HSV-1 (Herpes simplex virus 1), interferons (IFNs), antigen-presenting cells (APCs), dendritic cells (DCs). Created with BioRender.com.

**Table 1 ijms-24-13215-t001:** Principal cells of the tumour microenvironment of gliomas.

Cell Type	Function within the Tumour Microenvironment (TME)	References
Glioma cells	Secrete immunosuppressive cytokines Downregulate major histocompatibility complex (MHC) class I expression Upregulate programmed death-ligand 1 (PD-L1) expression Remodel the extracellular matrix Release growth factors that promote angiogenesis, proliferation, invasion and immune evasion	[13,14,15]
Tumour-associated macrophages and microglia (TAMs)	Mostly M2 phenotype promoting glioma growth and immune suppression Release interleukin 10 (IL-10), tumour growth factor beta (TGF-β) and IL-12 Suppress T-cell and NK-cell activity	[13,14,15]
Regulatory T (T_reg_) cells	Inhibit effector T-cell activity and promote immune evasion Increase cytotoxic T-lymphocyte-associated protein 4 (CTLA-4) and programme cell death protein 1 (PD-1) expression, suppressing anti-tumour pathways	[13,14,15]
Natural killer (NK) cells	Recognise and kill glioma cells Produce interferon gamma (IFN-γ), tumour necrosis factor alpha (TNF-α) and IL-12, promoting anti-tumour immune responses	[13,14,15]
Dendritic cells (DCs)	Antigen-presenting cells (APCs) that can active T cells and initiate anti-tumour immune response	[13,14,15]
Myeloid-derived suppressor cells (MDSCs)	Immunosuppressive cells that inhibit the activity of T cells and NK cells, promoting immune evasion	[13,14,15]

**Table 2 ijms-24-13215-t002:** Recently developed immunotherapies for glioblastoma.

Immunotherapy	Description	References
Immune checkpoint inhibitors (ICI)	Monoclonal antibodies that block either the programmed cell death protein 1 (PD-1) or cytotoxic T-lymphocyte-associated protein 4 (CTLA-4) pathways, resulting in the activation of T cells to target cancer cells	[35]
Therapeutic cancer vaccines	Immunogenic agents designed to stimulate antigen presentation and immune activation against cancer cells	[36]
Chimeric antigen receptors (CAR) T-cell therapies	T cells are genetically engineered to express CAR that can recognise specific tumour antigens	[37]
Oncolytic virotherapy (OVT)	Engineered viruses selectively infect and kill cancer cells, inducing an immune response against tumour antigens	[38]

## Data Availability

No new data were created or analyzed in this study. Data sharing is not applicable to this article.

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
