# Peer review of "Hallmarks of the Tumour Microenvironment of Gliomas and Its Interaction with Emerging Immunotherapy Modalities"

_ijms, 2023, doi:10.3390/ijms241713215_

Round 1
Reviewer 1 Report

Suitable
Author Response
Dear Editor and Reviewers,
I am pleased to resubmit for publication the revised version of TAM-23-03-193 manuscript, entitled “Emerging hallmarks of the Tumour Microenvironment of Gliomas and the Interplay with Immunotherapy”.
Thankfully the reviewers provided us with a great deal of guidance, regarding how to better position the article. We are hopeful you agree that this revision will update our invited editorial. All the comments have been addressed, as shown in the revised version of the manuscript, along with this point-by-point response to the reviewers' comments.
All corresponding are blue changes in the manuscript.
Reviewer 1:
Global remark:
Not all of the review is as fluid as it could be, except for the OnvoVirus part, which is perfect for me. In fact, if left unchanged, it looks more like a reader's digest, with a lack of personal involvement in writing, of some other already good reviews on the subject, rather than an original way of summarizing recent discoveries and relating them to a specific question, as the title suggests.
You should go more on details while describing the referenced studies and stay away from the other reviews you read to build this one.
You should go into more detail when describing the referenced studies and what they add, and stay away from the other reviews you read to build this one.
The part on oncoviruses is well written and more original; I encourage you to write more in the same way and to open the perspectives to other forms of unconventional therapy currently being evaluated in cancers, which generate an immunogenic death quite suitable for future combinations in immunotherapy.
Response: After making content corrections, the entire manuscript was reviewed for voice uniformity.
Detailed reviewing:
Intro
Provide more information on its importance in the world (map, like the one in GLOBOCAN eg..)
Response: Reformulated the introduction to better set the context, but using CRUK data instead of GLOBOCAN
Part I
Ia.
-
Not all myeloid cells are immunosuppressive, you need to describe the different subtypes.
-
complete table 1 accordingly + add neutrophils and add a new column with references
Response: Clarified in text; Table 1 refers only to the microglial TAMs
Microglie and TAMs
-
Hot and Cold tumor do not refer to the phenotypical status but rather the infiltration state (quantitity and quality). ïƒ to better explain
-
Be more precise about what bring the ref 17 about M2 macrophages and GB
Response: Clarified the hot/cold metaphor throughout the text; Lu-Emerson et al. 2013 (reference 17 in the original submission) mentions M1, which exhibits a proinflammatory response, and M2, which exhibits an immunosuppressive phenotype
DC
-
Differentiate the different status of DC development (immature vs. mature) and their immunological role for a better comprehension
-
Better description of the trials about DCs with key features is required
Response: Done
Ib.
-
The ref 19 do not refer the TME as Th2 … you have to precise your thought and be more accurate about what you want to explain.
-
The 19 refers instead to a dysfunction of the T lymphocytes which it would be useful to recapitulate in order to have a better vision of the T lymphocyte compartment in the WBC.
Response: Done
Treg
Regulatory T cell OR Regulatory Lymphocyte but not both as you wrote: Regulatory T cell lymphocyte
Response: Done
Make more link between your sentences and the information you give. Otherwise, the written word resembles a patchwork of unrelated ideas, an accumulation of sentences.
Response: After content corrections, the manuscript voice has been revised for better uniformity and coherence.
At the end of the paragraph, we do not understand very well what it means. Please reformulate, be more precise and make links. Please do not refer to a review to explain a research program but cite the referenced article itself and explain more deeply: you cite 19 and 26 but in the 19 they already refer to 26…
Response: Done
NK
Be careful!!! NK are CD3 NEGATIVE per se!!!! Please correct this. Exemplify KIR and KAR at least …
Response: Done
Link between HLA-G, NK cells and suppression? Please explain
Response: Done
Ic.
CTL: described first in full letter before abbreviate.
Response: Corrected
Explain Exhaustion whicc is an important notion that has bring the IO to what is now and will help further studies to develop.
Response: Expanded in the text
Transforming TGFb?
The last sentence of the last paragraph needs to be rewritten in a better conclusive way.
Response: Corrected, and the paragraph reformulated in a better attempt to describe the evasion.
Part II
IIa.
For the Dexamethasone part :
As per my previous remark, about 19… please do not refer to this review if you explain the published work and conclusion of another team… here mention, after explaining more deeply, the work of Badie et al. for example…
-
Alongside 2*repetitions to suppress
-
Cancer primariesïƒ primary cancers
The end of the paragraph needs to be rewritten since it’s the accumulation of sentences without link. Please use connecting words.
Response: This paragraph has been reformulated for clarity, noting the two corrections and referencing Badie’s 2000 paper on dexamethasone.
Table 2: ICI do not only refer to antiPD1 or anti-CTLA4, please give a more wide description, what you wrote are examples not a definition… It’s the same for all other entries of the Table… Please change it into a 4 columns table with: Immunotherapy /Definition/Example/reference (in GB)
I am not agree with the definition of ICI depicted here since it is restricted to the PD1 or CTLA4 axis only. It would be better to introduce this paragraph explain that nowadays only 3 approches has gain approvals, while many others are in current clinical trials in many cancers (certainly in GB too).
Response: Corrected by quickly narrowing the focus to the ICIs that are currently approved for clinical use; limited by word count and scope
In Figure 1: avoid naming the moleculs (Nivo, Ipi etc.) and just let the target (anti-PD1, antiCTLA4…) or cite all the molecules …
Response: Done
-
Please also discuss about approaches in clinical trials for other IC targets (antiTIM3, antiLAG3…)
Figure 1 : in the vaccine part, please change the sentence describing DC as able to “recognize Tumor cells” which is incorrect. Only T cells educated by trained DC will…
Response: Added a bit mentioning IC targets on the horizon such as anti-TIM3 and anti-LAG3 (Yang K 2022 Glioma targeted therapy: insight into future of molecular approaches | Molecular Cancer | Full Text (biomedcentral.com)
IId.
Please described again CARs in the beginning of the introducing paragraph and cite your figure 2a.
Response: Done
Please explain in which way the PD1-DNR is promising, what are the result of the study what difference compared to anti-PD1 combination.
Response: Done
The last paragraph for CAR-T cell part, about Bispecific is not at the right place and rather belong to ICI. Please advise.
Response: Reviewed and corrected
For CAR-NK, rather than saying “in an MHC-independent” mechanism you‘d better write in an antigen independent, since, as you explain elsewhere, recognize MHC or MHC like molecules…
Response: Done
Make one sentence while talking about other NK cytotoxic mechanisms. IIe.
Response: Done
Excellent part of the Review, all the other ones needs to be re-written the same way.
Response: Many thanks, have attempted to unify the voice
Conclusion
I can’t read in this conclusion about Emerging HallMarks in GB usfull for Immunotherapy. Please advise.
Response: Have reformulated the conclusion altogether, after accounting for all the corrections and content updates
Other remarks:
-
Write a few words in each paragraph of the second part about the ADVERSE EFFECTS: side effects of immunotherapies. Also, remember that many patients do not respond (primary resistance) and that some of those who do respond relapse (secondary resistance), then introduce this limitating statement.
Response: Done, adverse events and/or safety mentioned for each category
-
Introduce other types of new therapies to gain in originality such as photo-dynamic therapy, which leads to immune activation and is currently booming in the treatment of cancers.
Response: Sincere apologies, but unfortunately PDT is beyond the scope of our review and would require an extensive discussion comprising also inorganic nanoparticles, magnetic hyperthermia and perhaps others, and we are already quite limited after incorporating extensive feedback from three reviewers.
-
I'm not convinced by the title, because it doesn't match the writing and even the conclusion doesn't answer it...
Response: Revised as per specific suggestion by Reviewer 3
Reviewer 2 Report
File in attach

· In Tab. 1 please correct “natural kills cells”.
Author Response
Dear Editor and Reviewers,
I am pleased to resubmit for publication the revised version of TAM-23-03-193 manuscript, entitled “Emerging hallmarks of the Tumour Microenvironment of Gliomas and the Interplay with Immunotherapy”.
Thankfully the reviewers provided us with a great deal of guidance, regarding how to better position the article. We are hopeful you agree that this revision will update our invited editorial. All the comments have been addressed, as shown in the revised version of the manuscript, along with this point-by-point response to the reviewers' comments.
All corresponding are blue changes in the manuscript.
Reviewer 2:
The review article entitled “Emerging hallmarks of the Tumour Microenvironment of Gliomas and the Interplay with Immunotherapy” by C. A. Linares & A. Varghese and colleagues is a comprehensive
report for the benefit of the entire scientific community studying brain tumors and glioblastoma in particular. It is important as a review of the latest advances in the field. In the literature there are so many data, thus this work facilitates an overview of immunotherapy applied to glioblastoma.
Nevertheless, the text needs some major revisions:
-
Make the title more fluid and thus more appealing.
Response: Revised as per specific suggestion by Reviewer 3
-
Chapter “Hallmarks of the Tumour Microenvironment of Gliomas” includes many information that are not related to the immunotherapy. Please, integrate only the information about the targets of the therapies on which the study focuses into the relevant paragraphs to bring the reader immediately to the highlight of the review paper.
Response: Attempted to simplify, leading more quickly to the chapter on immunotherapies and their interaction with the TME
-
In Tab. 1 and Tab. 2 specific references need to be added for all details reported
Response: Done
-
TAMs categorization in two phenotypes is incorrect (pag.6, lane 5), please correct.
Response: Corrected
-
Tab.2: specify target cells, mechanisms of action, model, or clinical phase for each immunotherapy.
Response: Targets mentioned in the descriptions for each type of immunotherapy
-
The paragraph “Immunotherapy and the interplay” has an unclear title and the subparagraph “Immunotherapy landscape in glioma” needs to become the introduction of the paragraph.
Response: Revised
-
Add a figure for cancer vaccines.
Response: Done
-
For lots of abbreviations the text is very difficult to be fluently read. Please, use a reduced number of abbreviations.
Response: Removed all abbreviations which are non-standard, not repeated or otherwise don’t help with word count, especially common words such as glioblastoma, radiotherapy and chemotherapy
-
Figure 1: caption must describe the entire figure and details need to be reported in the figure.
Response: Done
-
Pag.12, lane 21, the refence to Figure 1 is inappropriate because the difference between “preventive” vs “therapeutic” is not reported in the figure. Please eliminate this reference or add this difference to the figure.
Response: Corrected
-
Figure 2: details reported in the caption are not reported in the text. Please, correct.
Response: Removed from the caption
-
Pag. 17, lane 3-4: pros and cons of each platform and the different modes of delivery must be better described and discussed.
Response: Sincere apologies, but we found that attempting to add this large additional information would greatly skew the balance and scope of our review, and we are already well over the word count.
-
Figure 3: panels A and B are not reported in the caption.
Response: Corrected
Minor appointments:
-
In the Introduction:
-
Lane 3, Authors wrongly refer to “other paediatric cancers”, please correct.
-
Lane 6, “originating from various glial cell lines”, please clarify or correct.
-
Clarify the difference between glioma and glioblastoma in this section.
-
Response: Corrected
-
Title in paragraph (a) is misleading because BBB is made up of cells.
Response: Updated
-
A reference about the % of microglia (“10-20% of the non-neuronal cells”) is needed.
Response: Removed
-
Which cells release CCL2? Please, clarify.
Response: Glioma cells themselves secrete CCL2, clarified in the text
-
Clarify “hot” versus “cold” TEM.
Response: Metaphor clarified throughout
-
Pag. 6, lane 26-28 and pag. 7, lane 1-5: clarify the “substance entry” through the BBB, which type of tumors compared to glioma show the increased immunosurveillance, which intravenous treatment shows a poor efficacy; the link between low blood flow and the macrophage attraction.
Response: Clarified to indicate restriction of leukocyte entry, lipophobic restriction of BBB and tumourigenicity enhancement
-
Many important features of NK cell role in GBM progression are lacking.
Response: Added the roles of KIR, KLR and KAR
-
In paragraph (e) Authors refer to “different types of cancer cells” that are not mentioned before. Please add this information.
Response: Removed
-
Pag. 7, lane 7, use “physiologically” or “in homeostatic condition” instead of “naturally”
Response: Corrected
-
Pag.7, lane 18, please clarify “in another context”
Response: Corrected
-
The title “immunotherapy and the interplay” is not definite.
Response: Revised
-
Pag. 10, reference 32,33 are not detailed or discussed
Response: References removed; appears we had cut some content during editing
Reviewer 3 Report
The review article, titled "Emerging Hallmarks of the Tumor Microenvironment of Gliomas and the Interplay with Immunotherapy" by Christian A Linares, provides a comprehensive overview of the hallmarks of the glioma TME and its interaction with emerging immunotherapy modalities. Immunotherapy has emerged as a promising approach to overcome these challenges, with various modalities such as immune checkpoint inhibitors, cancer vaccines, oncolytic viruses, CAR T cells, and NK cells showing potential. Unfortunately, to date, gliomas still lack effective therapies, and therefore any advance in the treatment is of the utmost importance.
Overall, the manuscript presents a comprehensive analysis of the topic. However, I have several minor comments and suggestions to improve the manuscript:
1. The topic is timely, and the manuscript is informative, equipped with interactive figures, and provides readers with information on the current state of the field. However, the manuscript fails to offer a critical view of several key issues within the field.
2. The manuscript should shed sufficient light on why the results from clinical trials of some of the relevant strategies fell short of expectations.
3. To strengthen the overall interpretation of the clinical outcomes, it is important to discuss potential biases, confounding factors, and limitations of the reviewed clinical trials. By acknowledging their limitations and considering their impact on the findings, the article will provide a more balanced perspective.
4. I suggest incorporating the following papers into the manuscript to further enrich the discussion: PMID: 36717668; 36376563; 36719372; 35534356; 37414817; 37067922; 36759557; 34586841; 35750880.
5. It would be beneficial to address the role of Dexamethasone in the success of immunotherapy and discuss why clinical trials for GBM have largely failed. Exploring the major limitations encountered in these clinical trials and discussing future directions to resolve such issues will provide valuable insights for the readers.
6. The manuscript lacks a chapter titled "Challenges and Future Directions," making it difficult for readers to identify the key take-home messages. I recommend including this new section in the revised version.
Author Response
Dear Editor and Reviewers,
I am pleased to resubmit for publication the revised version of TAM-23-03-193 manuscript, entitled “Emerging hallmarks of the Tumour Microenvironment of Gliomas and the Interplay with Immunotherapy”.
Thankfully the reviewers provided us with a great deal of guidance, regarding how to better position the article. We are hopeful you agree that this revision will update our invited editorial. All the comments have been addressed, as shown in the revised version of the manuscript, along with this point-by-point response to the reviewers' comments.
All corresponding are blue changes in the manuscript.
Reviewer 3:
The review article, titled "Emerging Hallmarks of the Tumor Microenvironment of Gliomas and the Interplay with Immunotherapy" by Christian A Linares, provides a comprehensive overview of the hallmarks of the glioma TME and its interaction with emerging immunotherapy modalities. Immunotherapy has emerged as a promising approach to overcome these challenges, with various modalities such as immune checkpoint inhibitors, cancer vaccines, oncolytic viruses, CAR T cells, and NK cells showing potential. Unfortunately, to date, gliomas still lack effective therapies, and therefore any advance in the treatment is of the utmost importance.
Overall, the manuscript presents a comprehensive analysis of the topic. However, I have several minor comments and suggestions to improve the manuscript:
1. The topic is timely, and the manuscript is informative, equipped with interactive figures, and provides readers with information on the current state of the field. However, the manuscript fails to offer a critical view of several key issues within the field.
Response: With the comprehensive feedback below and from the other 2 reviewers, we feel this has been well rectified now.
2. The manuscript should shed sufficient light on why the results from clinical trials of some of the relevant strategies fell short of expectations.
Response: All referenced trials were reviewed to ensure some critical comment where possible/significant to the narrative.
3. To strengthen the overall interpretation of the clinical outcomes, it is important to discuss potential biases, confounding factors, and limitations of the reviewed clinical trials. By acknowledging their limitations and considering their impact on the findings, the article will provide a more balanced perspective.
Response: All referenced trials were reviewed to ensure some critical comment where possible/significant to the narrative.
4. I suggest incorporating the following papers into the manuscript to further enrich the discussion: PMID: 36717668; 36376563; 36719372; 35534356; 37414817; 37067922; 36759557; 34586841; 35750880.
Response: Many thanks for pointing out these important papers. We were able to incorporate several of them.
5. It would be beneficial to address the role of Dexamethasone in the success of immunotherapy and discuss why clinical trials for GBM have largely failed. Exploring the major limitations encountered in these clinical trials and discussing future directions to resolve such issues will provide valuable insights for the readers.
Response: Have now discussed Badie et al regarding dexamethasone
6. The manuscript lacks a chapter titled "Challenges and Future Directions," making it difficult for readers to identify the key take-home messages. I recommend including this new section in the revised version.
Response: Incorporated through the text instead of as a separate section, as quite limited by word count after all other additional content requested by 3 reviewers
Round 2
Reviewer 3 Report
In reviewing the manuscript, I noticed that there appears to be a missing citation in the reference section. In the text, at point [section future direction where Citation 95 is mentioned], the authors refer to Citation 95 to support their argument on oncostreams and Col1A1. However, upon checking the reference list, I couldn't find an entry corresponding to Citation 95. I kindly request the authors to ensure that all citations mentioned in the text are accurately listed in the reference section.
Author Response
Dear Reviewer,
Thank you very much for your consideration.
We have double-checked all the references. We had to maintain reference 95, but edited its styling according to journal's requirements.